# Increased alertness and moderate ingroup cohesion in bonobos' response to outgroup cues

James Brooks[1,2,3]*, Karlijn van Heijst[1], Amanda Epping[4], Seok Hwan Lee[2], Aslihan Niksarli[5], Amy Pope[6], Zanna Clay[7], Mariska E. Kret[8,9], Jared Taglialatela[4,10], Shinya Yamamoto[1,2]

1 Institute for Advanced Study, Kyoto University, Kyoto, Japan, 2 Wildlife Research Center, Kyoto University, Kyoto, Japan, 3 Kumamoto Sanctuary, Kyoto University, Kyoto, Japan, 4 Ape Cognition and Conservation Initiative, Des Moines, Iowa, United States of America, 5 Department of Anthropology, Yeditepe University, İstanbul, Türkiye, 6 Department of Psychology, University of Warwick, Coventry, United Kingdom, 7 Department of Psychology, Durham University, Durham, United Kingdom, 8 Department of Cognitive Psychology, Institute of Psychology, Leiden University, Leiden, Netherlands, 9 Leiden Institute for Brain and Cognition (LIBC), Leiden University, Leiden, Netherlands, 10 Department of Ecology, Evolution, and Organismal Biology, Kennesaw State University, Kennesaw, Georgia, United States of America

* jamesgerardbrooks@gmail.com

**Data Availability Statement:** All relevant data are within the paper and its Supporting Information files.

## Abstract

In a number of species, including humans, perceived outgroup threat can promote ingroup cohesion. However, the distribution and selection history of this association across species with varied intergroup relations remains unclear. Using a sample of 8 captive groups (N = 43 individuals), we here tested whether bonobos, like chimpanzees, show more affiliative ingroup behaviour following perception of outgroup cues (unfamiliar male long-distance vocalisations). We used comparable methods to our previous study of captive chimpanzees, and found that, although weaker, there was an association for more frequent social grooming in response to the outgroup condition than the control condition, alongside more alert posture and increased self-directed behaviour. This provides preliminary evidence for an ancestral origin to the proximate association between outgroup cues and ingroup cohesion, at least prior to the *Pan-Homo* split, and suggests the presence of intergroup competition in our last common ancestor.

## 1. Introduction

In a variety of species, including humans, perceived outgroup threat elicits ingroup cohesion. However, no direct comparative studies between species have been conducted, which are necessary for assessing the relevance of socio-ecological factors. Theoretical perspectives predict an association between intergroup competition and group cohesion [1–5], suggesting this association may be stronger in species which naturally face stronger intergroup competition. Here, we studied the impact of perception of outgroup cues on ingroup cohesion in bonobos (*Pan paniscus*), who, despite their close phylogenetic relation to both humans and

**Funding:** This research was financially supported by the Leading Graduate Program in Primatology and Wildlife Science of Kyoto University, and Japan Society for the Promotion of Science (21J21123 to JB, 19H00629 and 22H04451 to SY) and Japan Science and Technology Agency Fusion Oriented REsearch for disruptive Science and Technology (JPMJFR221I to SY). The funders had no role in study design, data collection and analysis, decision to publish, or preparation of the manuscript.

**Competing interests:** The authors have declared that no competing interests exist.

chimpanzees (*P. troglodytes*), lack lethal intergroup aggression. We followed the protocol and observation methods of Brooks et al. [6], which found perception of outgroup cues can promote ingroup cohesion in captive chimpanzees, in order to directly compare the impact of hearing the vocalisations of unfamiliar outgroup members on ingroup social behaviour between two sister species differing in natural intergroup relations.

While the association between outgroup threat and ingroup cohesion has long been known in humans, only more recently have empirical studies examined the effect across other species. Of these, the majority have focused on cooperative breeders (e.g. green woodhoopoes [*Phoeniculus purpureus*], dwarf mongooses [*Helogale parvula*], cichlid fish [*Neolamprologus pulcher*]), which find that across a range of taxa groups tend to unite together when presented with signs of an outgroup, at least in the short term [7–13]. However, the form and function of group-living and the social and ecological contexts of group relations varies, and there have been few studies looking at this effect in other grouping and breeding structures. Of them, studies targeting primate species have found varied results including both increases and decreases in ingroup cohesion after perceiving outgroup cues [14–18]. However, each has focused on one species at a time, and each employs differing methods, meaning the role of grouping structure and wild intergroup relations could not be directly assessed. Two comparative phylogenetic analyses of primate species on broader measures of group cohesion have been conducted, with one showing no association between intensity of intergroup competition and absolute grooming frequency [19] and another (focused primarily on Afro-Eurasian monkeys) finding an effect of intergroup behaviour only on female grooming density [20]. Neither of these studies addressed the proximate changes within species as driven by outgroup cues, but rather the long-term behavioural patterns of each species.

In chimpanzees, one of humans' two closest living relatives, studies in both wild [17] and captive [6] contexts have found increased group cohesion when presented with proximate outgroup cues. These studies provide evidence that the association is shared between humans and chimpanzees, but do not indicate whether it arose prior to our divergence or emerged independently in each species. While human intergroup relations cover the full spectrum from warfare to peaceful cooperation, intergroup relations in wild chimpanzees are universally hostile and involve both lethal aggression as well as group-level cooperation in territory defence [17, 21]. To date, there is little data to suggest whether the tendency to "unite against a common enemy" is conserved across the *Pan-Homo* lineage or whether it crucially depends on intense intergroup competition in more recent evolutionary history. Direct comparison to bonobos, our other closest relative, can test between these alternatives. If the behavioral phenotype is present in all 3 species of great ape, that would support the hypothesis that ingroup cohesion following perceived outgroup cues is an ancestral characteristic in the Pan-Homo lineage.

Bonobos diverged just 1–2 million years ago from chimpanzees, with both species diverging from humans around 5–6 million years ago [22–24]. Compared to chimpanzees, wild bonobos have significantly more tolerant intergroup relations overall [25, 26]. While intergroup competition does exist [27] and coalitionary aggression by males against outgroup members has been observed [28], lethal aggression has not been reported and intergroup encounters can involve cross-group grooming, play, and sexual behaviour, as well as multi-day foraging associations [25, 29]. Given these apparently striking differences in inter-group relations among chimpanzees and bonobos, we thus aimed to directly compare the response to perception of outgroup cues in bonobos to that reported in chimpanzees [6]. The presence of an association between outgroup cues and ingroup cohesion in bonobos would provide some evidence to distinguish between hypotheses on the emergence of this association in the human lineage. We formed two ultimate hypotheses for the evolutionary link between outgroup cues and ingroup

cohesion in bonobos, each with a proximate hypothesis for the direct impact of hearing unfamiliar outgroup vocalizations on bonobo behaviour.

The first ultimate hypothesis, the 'conserved association' hypothesis, suggests that a proximate association between outgroup perception and ingroup cohesion is conserved across *Homo* and *Pan*, and that it does not depend on actual recent history of intense intergroup competition. This hypothesis suggests that the phenomenon emerged at some point prior to the *Pan-Homo* split and can be found in bonobos despite not being under active selection in the modern socioecology. This therefore gave rise to the social cohesion hypothesis for the proximate effects of outgroup vocalizations on ingroup behaviour (developed in Brooks et al., suggesting that bonobo groups will become more socially cohesive in the outgroup condition compared to the control condition. The social cohesion hypothesis more specifically predicts increased social cohesion through predictions: (a) closer spatial proximity, (b) increased affiliation in social grooming and social play, and (c) decreased aggression, alongside (d) increased vigilance and stress observed in rest, posture, and self-directed behaviour.

Alternatively, the second ultimate hypothesis was the 'recent intergroup competition' hypothesis. This suggests instead that the phenomenon depends crucially on recent intense intergroup competition in a species' environment of selection and therefore predicts that bonobos, with their more tolerant intergroup relations, would not show the same pattern as humans and chimpanzees. This would not be able to provide any indication of whether the association could be found in the common ancestor, but would either way suggest that the association likely depends on intergroup competition in more recent evolutionary history. On a proximate level, this ultimate hypothesis gives rise to the null hypothesis for outgroup stimuli's effect on ingroup cohesion (predictions a, b, and c above) but remains neutral on their effect on vigilance and stress behaviour (prediction d).

Employing the same design to allow a systematic species comparison, we therefore aimed to test the social cohesion hypothesis of Brooks et al. [6] regarding the proximate consequences of hearing outgroup long-distance vocalizations (bonobo high hoots), relative to control vocalizations (crow 'ka' calls), on ingroup social behaviour in captive bonobos. While captive apes may not be representative of wild populations in several important ways, presentation of outgroup stimuli has proven effective at modulating behaviour in a range of primate species in more carefully controlled contexts, and we therefore aimed to follow consistent methodologies as the prior study with captive chimpanzees. The social cohesion hypothesis predicted that individuals would become more affiliative and less aggressive within the group, as well as more vigilant overall, after being presented with the outgroup cues [6]. It thus predicted that, relative to the control condition, bonobos in the outgroup condition would show increased rates of social grooming, decreased aggression, and increased social play, alongside less rest, more self-directed behaviour, and a higher proportion of their rest sitting upright. The alternative, the null hypothesis, is that there would be no measurable change to social behaviour following the outgroup calls, relative to the control condition.

## 2. Methods

### 2.1 Participants

8 groups of captive bonobos at 5 sites (N = 43, 17 males and 26 females) participated in this study. Groups ranged in size from 3 to 10 individuals. In 3 groups (CZ, FZ1, FZ2) there was a dependent offspring (<2 years old) who was not included in data analysis (final N = 40). Participating sites were: Kumamoto Sanctuary (KS), Japan; Ape Cognition and Conservation Initiative (AI), USA; Twycross Zoo (TZ), UK; and Frankfurt Zoo (FZ) and Cologne Zoo (CZ), Germany. At KS, AI, TZ, and FZ, two bonobo groups are housed in adjacent enclosures with

visual and auditory access to one another. At AI and KS, group membership rotates in a simulated fission-fusion social system, while at TZ and FZ groups are kept mostly consistent, except for occasional individual movements for management reasons. These grouping patterns are similar to those of chimpanzees in Brooks et al.; in this prior study 3 groups lived in a simulated fission-fusion environment, while 2 groups had relatively fixed membership, also with some visual and auditory access between groups. One of the groups in Frankfurt was observed only for 3 trials due to observer availability. Only one group at TZ was included in this study. In all cases, as in Brooks et al. [6], subgroups were treated as separate groups in all analyses. Membership of each group was constant for all trials at all sites, except for one individual (Lopori) joining the focal group for one trial at TZ. Enclosure sizes were comparable or larger to those of the chimpanzees in [6]. The most significant difference in study population relative to the previous study was the sex structure of group, where in Brooks et al. [6] there were 15 individuals in 3 all-male groups, and 14 individuals in 2 single-male multi-female groups, while in the present study most groups were multi-male, multi-female, except for KS1 (n = 3), which was all-female, and AI1 (n = 3) and KS2 (n = 3), which were both single-female multi-male.

## 2.2 Ethical statement

Ethical approval and relevant permissions were obtained at all sites involved in this study. Ethical approval numbers were WRC-2022-KS (for KS), WRC-2022-023A (for collaborating zoos), and IACUC Animal Use Protocol # 210929–01 for AI. No major changes were made to the daily care routine of bonobos at any site for the purpose of this study, and as such were not food or water deprived at any time.

## 2.3 Data collection

Data was live coded and followed the observation methods of Brooks et al. [6]. No changes were made to the observational protocol between the two studies. In short, at two-minute scans, interindividual distances of all dyads (estimated into 4 categories: in contact, within arm's reach, <3m away, and >3m away) were recorded, alongside individual behaviour of each group member at the time of scanning. In addition, all occurrences of play, aggression, and sexual behaviour, along with identity of individuals involved, were recorded. Observers were consistent across all trials within groups, and observed only one group at a time (one observer worked at 2 sites; FZ and CZ, and all other observers observed one group each). All observers trained in pairs on-site for several practice days prior to experimental trials, until reaching a high level of confidence with consistency in data recording. Due to practical constraints, observers were not blind to the conditions and hypotheses being tested.

## 2.4 Experimental protocol

All procedures followed Brooks et al. [6], involving 30 minutes of observation before, during and after playback, the pre, playback, and post phases, respectively. The main difference from the prior study of chimpanzees was the absence of a food phase (when semi-monopolizable enrichment tubes were provided in the 30 minutes after sound presentation), which was replaced by a "post" phase during which no sounds or stimuli played. This change was due to the inherent variation across sites in enrichment strategies and limitations in caretaker time, where we instead observed the after-effects of stimuli presentation which could be performed comparably at all sites. Only the pre and during phases were observed at TZ due to constraints on timing of care routines. No additional food was provisioned during any observation period, and feedings were kept consistent between morning and afternoon sessions across all trial

days. Specifically, in some sites this entailed food being provided directly before the pre phase in both sessions meaning the pre phase was initially a feeding period (and thus not directly comparable to other phases or between sites).

Four days of experiments were run for each group (except for one group at FZ, where only three trials were recorded). Each day involved a morning and afternoon session. On each experimental day, one session was an outgroup trial, and the other control trial, with orders or which condition was in the morning following an ABBA design, which was then counterbalanced across groups. At sites where multiple groups were observed, all groups were presented with the same condition at the same time (due to being in auditory contact of one another).

## 2.5 Stimuli

Stimuli consisted of 12 unique long-distance high-hoot vocalizations of distinct unfamiliar male bonobos. All outgroup recordings used in this study were of bonobos living in DRC and thus were unknown to all study participants. High hoots were chosen as the closest analogy to chimpanzee pant hoot and used in long-distance communication [30, 31], though we did not formally select vocalizations based on the social context of their production. Control crow "ka" vocalizations were the same recordings as those used in Brooks et al. (itself following Kutsukake et al. [32]). It should be noted that due to the geographical distribution across sites, the resident corvid species (and therefore their vocalizations) differ. We did not adapt the control vocalizations to each site, so it is possible there is some variation in the absolute novelty of the control stimuli between groups. As in the prior study, four distinct recordings (of four distinct individuals) of 15 seconds each were played in a given trial (repeated or cut to exactly 15 seconds), each separated by one minute of silence. 15 minutes after the start of the first sound, the same four sounds were repeated. Stimuli were played at 80-90dB measured from 10m away (following Kutsukake et al., and played from speakers placed 10-20m away from the bonobo enclosures (due to some limitations on consistency across sites). The first three trials were all unique stimuli, while the last included 4 sounds randomly selected from the first 12 of that category (outgroup bonobo or crow), with the same sounds used in the final trial across all groups. Example stimuli can be found in S1 File.

## 2.6 Analysis

Analysis strategies followed Brooks et al. [6]. A CLMM was run on proximity data (with each scan for each dyad representing one data point) from R package ordinal [33]. The dependent variable was closeness between dyads in four ordinal categories; 1 (in contact), 2 (within arm's reach), 3 (<3m), and 4 (>3m away). Binomial GLMMs with logit link function were run on scan behaviours using the glmer function from package lme4 [34], with each scan for each individual representing one data point, as either showing or not showing the given behaviour. For posture, data was first restricted to the data where individuals were resting, and a binomial GLMM was run on whether the individual was sitting or lying down. Binomial GLMMs were also run on all occurrence behaviours, with each trial for each individual representing one data point, as either showing or not showing the given behaviour (in order not to inflate the data when the same individual(s) repeatedly show the same behaviour within a given trial).

All models were structured in similar ways: As fixed effects, they included condition (outgroup vs. control), trial (1–4, normalized to a mean of 0 and standard deviation of 1), and their interaction. As random effects, they included random slopes of the fixed effects as well as time since start of the phase (for scan behaviors and proximity, normalized) and time of day (morning or afternoon) on the random effect of individual (nested within group). The proximity data had the same random slopes with random effects dyad ID, individual 1, and individual 2

(all nested within group), where individual 1 and individual 2 were randomly assigned to include both individuals in the dyad. In all models where possible the bobyqa optimizer https://journals.plos.org/plosone/article?id=10.1371/journal.pone.0246869 - pone.0246869. ref033[35] was used. In the case of non-convergent models, we removed random effects in the following way (defined a priori in Brooks et al.): first, the random slopes for time of day then time since start of phase, then the nesting of individuals within group, then the random slope of the interaction between condition and trial, then the random slope of trial, then the random slope of condition. This sequence was chosen in order to retain the fixed effects whenever possible prioritizing the effect of condition as this was the main hypothesis to be tested in this study, and time since start of phase included more detail and is expected to have had a higher impact than time of day. The same structured simplification was carried out with singular models and the results below present convergent non-singular models, while the maximal singular models are available in S1 File. For the analyses below, we give the formula of the final converging non-singular model. To examine the impact of participant sex, admittedly as exploratory analyses not conducted in the chimpanzee study (due to absence of multi-male multi-female groups) the final proximity and behavioural models were also run with an additional interaction between participant sex (or dyad sex pairing) and condition.

Final model structure for the proximity data for the playback phase was: proximity ~ trial * condition + (condition+trial|dyad) + (condition+trial|id1) + (condition+trial|id2); and for the post phase was: proximity ~ trial * condition + (condition*trial|dyad) + (condition*trial|id1) + (condition*trial|id2). Final model structure for self-directed behaviour in the post phase, rest in both the playback and post phases, and sitting in both the playback and post phases was: behavior ~ trial * condition + (condition*trial|individual). Final model structure for grooming in the playback phase was: behavior ~ trial * condition + (condition+trial|individual). Final model structure for self-directed behaviour in the playback phase and grooming in the post phase was: behavior ~ trial * condition + (condition|individual). Final model structure for all models of all occurrence behavioural data was: behavior ~ trial * condition + (1|individual).

Significance was calculated using chi-squared likelihood ratio test with the drop1 function [34] which uses full—null model comparison for hypothesis testing and an alpha value of 0.05. Specifically, it compares the full model to one generated by dropping only the fixed effect or interaction of interest but keeping other fixed and random effects the same. If the interaction between condition and trial was not significant, the model was run again with the interaction term removed. This procedure was followed for all variables of interest in separate models for both the playback and food phases. We additionally calculated odds ratio (OR) estimates and 95% confidence intervals for all significant effects. For interactions, the odds ratio represents the odds of a one unit increase when both variables are present over and above the main effects. For example, if there is an interaction between trial and condition the interaction term represents how much the response changes with every trial in the outgroup condition over and above the change that is due to trial and condition alone.

## 3. Results

There was no change in proximity in either phase by condition (playback phase: $\chi^2 = 0.31$, p = 0.58; post phase: $\chi^2 = 0.54$, p = 0.46; Fig 1).

Fig 2 shows behavioural results. There was a significant interaction between condition and trial in rates of self-directed behaviour in the playback phase ($\beta = 0.44$, SE = 0.17, $\chi^2 = 7.09$, p = 0.0078; OR = 1.55 (95% CI: 1.12, 2.15), but no effect of condition in the post phase ($\chi^2 = 0.094$, p = 0.76). There was a significant interaction between condition and trial in rates of grooming in both phases (playback phase: $\beta = 1.21$, SE = 0.15, $\chi^2 = 76.60$, p < 0.001; OR = 3.36

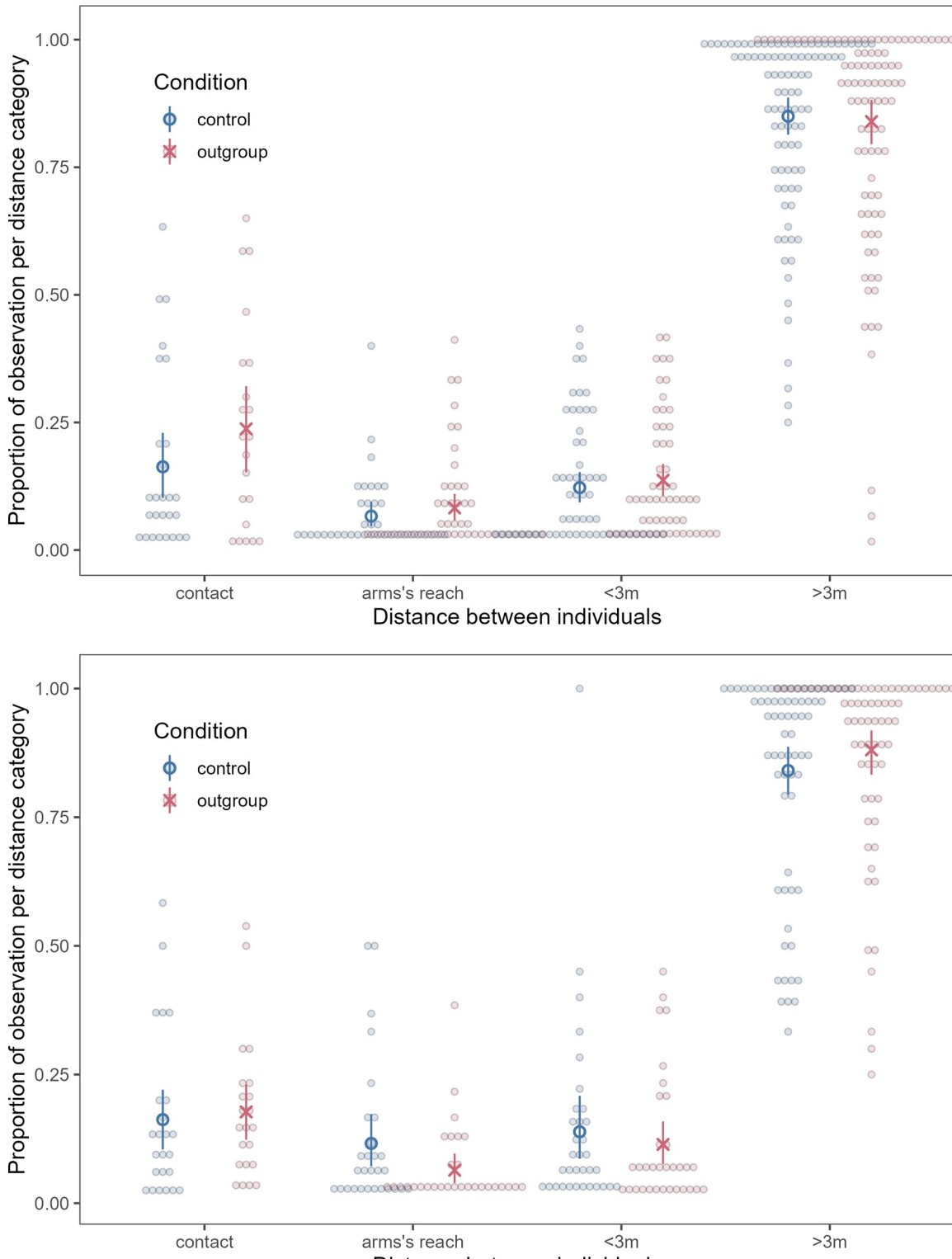

**Fig 1.** Proportion of observations in each proximity category in the outgroup and control conditions in a) the playback phase and b) the post phase. Each dot represents a dyad, blue circles represent the mean for the control condition, and red X's represent the mean for the outgroup condition. Red and blue bars represent 95% confidence intervals around the mean (based on a non-parametric bootstrap of the data).

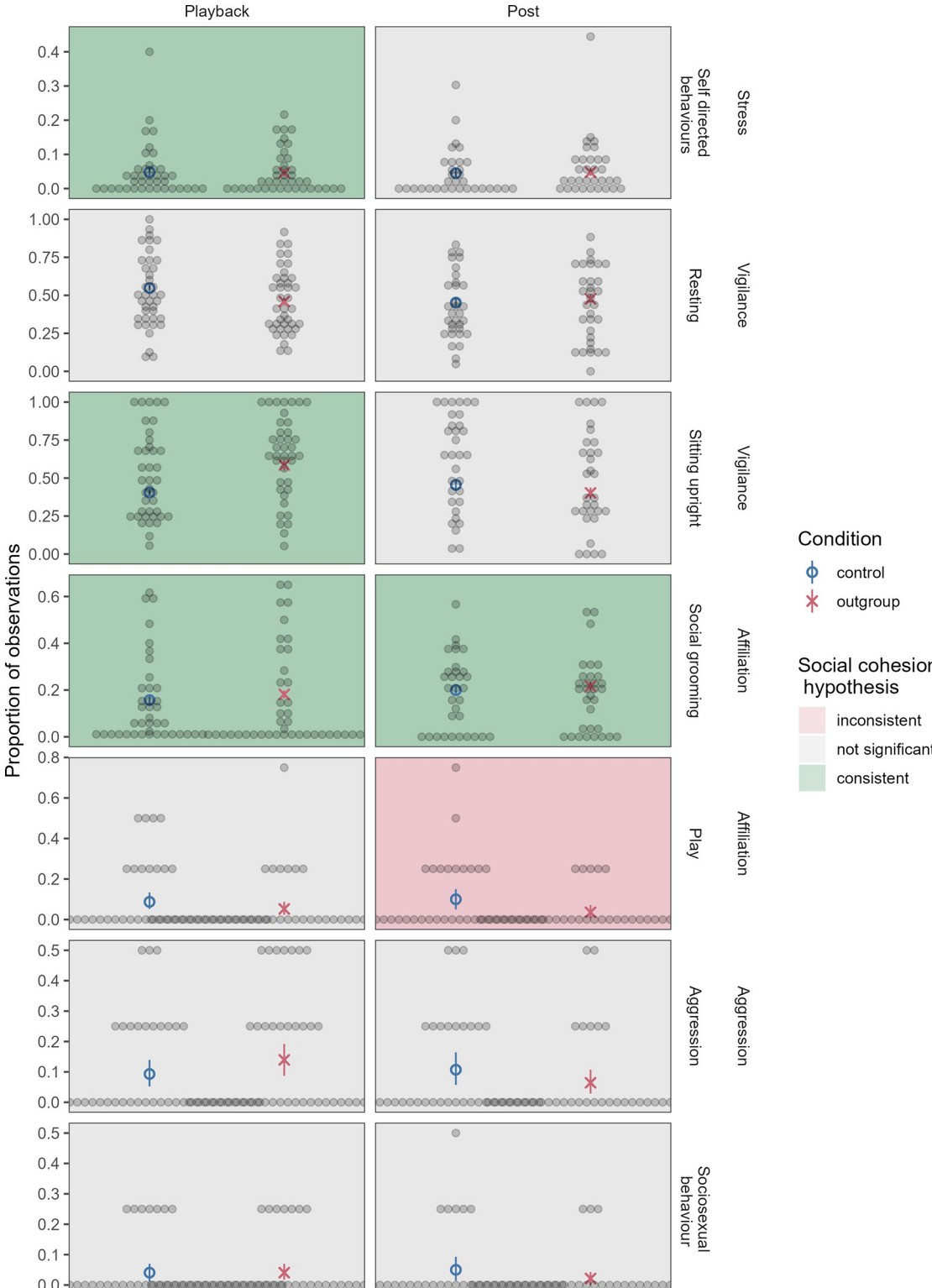

**Fig 2. Behaviors in the playback and post phases.** Self-directed behaviour in the playback phase, and social grooming in both the playback and post phase, had significant interactions between condition and trial, with the main effect of condition visualized here (graphs depicting trial interactions can be found in S1 File). Each dot represents an individual, blue circles represent the mean for the control condition, and red X's represent the mean for the outgroup condition. Red and blue bars represent 95% confidence intervals around the mean (based on a non-parametric bootstrap of the data).

(95% CI: 2.53, 4.47); post phase: $\beta$ = 0.30, SE = 0.099, $\chi^2$ = 9.29, p = 0.0023; OR = 1.35 (95% CI: 1.12, 1.64)). Interaction plots for significant interaction effects can be found in S1 File.

There was a non-significant marginal effect of condition on rest in the playback phase ($\beta$ = -0.29, SE = 0.17, $\chi^2$ = 2.88, p = 0.090; OR = 0.75 (95% CI: 0.54, 1.04)), but no effect in the post phase ($\chi^2$ = 0.29, p = 0.59). There was a significant main effect of condition on proportion of rest time sitting upright in the playback phase (more upright posture in the outgroup condition; $\beta$ = 0.93, SE = 0.27, $\chi^2$ = 9.57, p = 0.0020; OR = 2.55 (95% CI: 1.50, 4.32)) but only a marginal effect of condition in the post phase ($\beta$ = -0.80, SE = 0.48, $\chi^2$ = 2.72, p = 0.099; OR = 0.45 (95% CI: 0.18, 1.14)). There was no effect of condition on rates of aggression in either phase (playback phase: $\chi^2$ = 2.01, p = 0.16; post phase: $\chi^2$ = 1.89, p = 0.17). There was no effect of condition on rates of play in the playback phase ($\chi^2$ = 1.94, p = 0.16), but a significant interaction between condition and trial on rates of play in the post phase ($\beta$ = -1.44, SE = 0.63, $\chi^2$ = 5.87, p = 0.015; OR = 0.24 (95% CI: 0.068, 0.81)), and no effect of condition on rates of sexual behaviour in either phase (playback phase: $\chi^2$ = 0.00, p = 1; post phase:: $\chi^2$ = 1.71, p = 0.19).

We found no interactions between condition and sex (or dyad type for proximity data) on any measured behaviours in either phase (proximity: playback phase: $\chi^2$ = 0.22, p = 0.90; post phase: $\chi^2$ = 1.12, p = 0.57; self-directed behaviour: playback phase: $\chi^2$ = 2.46, p = 0.12; post phase: $\chi^2$ = 1.42, p = 0.23; grooming: playback phase: $\chi^2$ = 0.97, p = 0.32; post phase: $\chi^2$ = 0.14, p = 0.70; rest: playback phase: $\chi^2$ = 0.77, p = 0.78; post phase: $\chi^2$ = 1.20, p = 0.27; posture: playback phase: $\chi^2$ = 1.64, p = 0.20; post phase: $\chi^2$ = 0.67, p = 0.41; aggression: playback phase: $\chi^2$ = 0.22, p = 0.64; post phase: $\chi^2$ = 0.073, p = 0.79; play: playback phase: $\chi^2$ = 0.049, p = 0.82; post phase: $\chi^2$ = 1.15, p = 0.28; sexual behaviour: playback phase: $\chi^2$ = 0, p = 1; post phase: $\chi^2$ = 0.49, p = 0.48).

## 4. Discussion

Previously, we showed that, similar to humans, captive chimpanzees show enhanced ingroup cohesion in response to perception of outgroup cues [6]. In the current study, we found a similar behavioural pattern in captive bonobos, whereby hearing the vocalizations of outgroup bonobos resulted in enhanced affiliation (prediction b; in social grooming but not in play) as well as vigilance and stress (prediction d; in self-directed behaviour and a greater proportion of rest sitting upright, but not in overall rate of rest), but did not impact spatial proximity (prediction a) or aggression (prediction c). All significant effects (except the proportion of rest sitting upright) interacted with trial, suggesting the effect was not uniform or universal. The pattern of significant and nonsignificant effects was similar to that of the chimpanzees except for the lack of effect of condition on proximity or rate of rest, but estimated coefficients for all significant effects were generally low, suggesting the effect of outgroup perception is less pronounced in bonobos than in chimpanzees. While most behavioural changes were short-lived and disappeared in the post phase, only grooming remained significant through both phases (although notably with small effect size). Overall, these results generally support a moderate interpretation of the social cohesion hypothesis for the proximate effects of outgroup stimuli on bonobo ingroup social behaviour, with some evidence for greater affiliation but a weaker pattern than that observed in chimpanzees. This suggests that while some elements of the association between outgroup threat and ingroup cohesion may not crucially depend on sustained recent lethal intergroup competition, it does likely decrease in intensity as intergroup competition becomes less central to a species' socio-ecology, as has been suggested for bonobos [25]. Together, these findings suggest further that the association may be best-understood an adaptive response to intergroup competition that is slow to disappear within the Pan lineage (or can be sustained even under the relatively weak intergroup competition of extant bonobos)

making it a possible case of phylogenetic inertia (where ancestral phenomena are maintained despite losing selective advantage) [36].

While bonobos in the wild do not face as serious intergroup competition as chimpanzees, this study suggests they still may tend to become overall more socially cohesive when presented with cues of the outgroup. Both species also sat upright and engaged in self-directed behaviour more in the outgroup condition, consistent with increased vigilance, though in the case of bonobos (where intergroup encounters can be affiliative) these could also be interpreted as social anticipation rather than vigilance per se. In any case, bonobos, like chimpanzees, were more alert in the outgroup condition.

However, not all results we found in chimpanzees replicated to bonobos, despite a highly similar design. In the playback phase (the phase for which direct species comparisons are meaningful), there was no effect of condition on spatial proximity or on rates of rest. As with chimpanzees, several effects in the playback phase were interactions with trial (in chimpanzees: rest, posture, social grooming alongside main effect of self-directed behaviour, in bonobos: self-directed behaviour and social grooming alongside main effect of posture). However, when examining the change across trials, the pattern in bonobos is more suggestive of sensitization than habituation (see S1 File). This pattern was consistent across all the measures which had a significant interaction with trial, and opposite to the pattern that seemed more indicative of habituation found in chimpanzees [6]. At present, the reason for this difference, and how the pattern may change after more trials, remains speculative. Differences between sites make it difficult to interpret whether this opposing pattern is a real species difference or due to other factors that differed between the populations that were studied (such as group compositions, presence of juveniles and infants, and rearing histories). If the same species difference can be replicated in other groups, it would indicate an unexpected finding warranting more direct future study, but given the present results any interpretations should remain cautious.

There were no effects on all occurrence data in bonobos, except for play in the post phase, where rates of play in this phase decreased relative to the control condition. This was contrary to the social cohesion hypothesis, though consistent with the decreased play described in the playback phase in chimpanzees (possibly a consequence of increased vigilance). Besides this decrease in play, the only all occurrence effects in the chimpanzee study were found in the food phase, so despite lack of effects the pattern is similar given that no food was presented to bonobos. Rates of sex remained low across conditions and phases in bonobos, making direct comparisons between conditions difficult. Still, it is noteworthy that rates of sexual behaviour did not increase given the importance of GG-rubbing reported in wild intergroup encounters [25]. There are several possibilities for this lack of effect, including that the experimental design was not suited to bring about effects on rates of socio-sexual behaviour (we used only auditory stimuli which may be less salient than visual or olfactory cues), that the sex of the vocalization producer affects behavioural response (we here used male vocalizations to be consistent with the chimpanzee study, discussed further below), or that the spaced out and individual sound recordings were not salient enough to elicit the social tension which may be relieved by sexual behaviour.

Of future studies, hormonal work can help answer some of the most immediate questions and will be necessary to help reveal the specific physiological mechanisms by which outgroup cues induce ingroup cohesion. A bonobo study following the methods of Kutsukake et al. [32], which found outgroup cues caused higher salivary cortisol alongside increased vigilance in chimpanzees will be an important and relatively straightforward species comparison. Cheng et al. [27] reported increased urinary cortisol, a hormonal indictor of stress, during intergroup encounters in wild bonobos, consistent with earlier findings from chimpanzees [37], suggesting similarity between the species. The oxytocinergic system, further has been implicated in

preparation for and engagement in intergroup encounters in wild chimpanzees [37, 38], out-group social attention in both bonobos and chimpanzees [39], and social grooming in both bonobos [40] and chimpanzees [41]. Given oxytocin's importance in both intergroup competition and ingroup cohesion [42], it may form an important part of the proximate hormonal mechanism responsible for eliciting short-term ingroup cohesion in response to outgroup cues. A crucial remaining question in the case of bonobos is whether urinary oxytocin is related to intergroup behaviour in the wild. Notably, Wirobski et al. [43] found similar behavioural effects, supported by different hormonal mechanisms, in the territorial behaviour of wolves compared to dogs. In wolves, but not dogs, urinary oxytocin correlated with territorial behaviour (ground scratching, patrolling along the fence, simultaneous marking) and synchronized locomotion, while the absolute rates of these behaviours did not differ between the species. It thus remains possible that the effects described here and in Brooks et al. [6], while behaviourally similar, may be supported by different hormonal mechanisms. Wild studies following Samuni et al. [38] will be necessary to reveal whether bonobo intergroup behaviour in natural contexts is supported by the oxytocinergic system.

Sex differences are another clear future direction especially in light of sex differences found in Samuni et al. [17] in wild chimpanzees and Mirville et al. [15] in wild mountain gorillas. While we did not detect any differences between the responses of male and female bonobos, this should not be considered evidence against the existence of sex effects but rather inconclusive, given the low sample sizes and limited naturalism of this study. Bonobos and chimpanzees differ not only in intensity of intergroup competition, but also in the sex which is more involved in intergroup encounters (in chimpanzees males predominantly engage in intergroup encounters, though with varied female involvement across subspecies [21], while in bonobos females predominantly engage in between-group associations [25]). Some sex difference between species in response to intergroup cues may thus be expected, and should be a target of future studies. Similarly, with respect to stimuli, while we here aimed to maximize consistency with the prior study of chimpanzees, in the wild female bonobos play an important role in the initiation of group encounters and previous eye-tracking research has found similar attentional biases in bonobos' response to female outgroup stimuli as chimpanzees' response to male outgroup stimuli [39, 44]. Another study found attentional bias to emotional outgroup compared to ingroup stimuli in bonobos [45], interpreted as interest in potential interaction with outgroup members, suggesting the potential for a complex interaction between participant sex, stimulus sex, and emotional setting that could not be targeted here. It remains possible that female long-distance vocalizations would be a more ecologically relevant stimulus to simulate bonobo intergroup encounters.

The increase in grooming in both captive bonobos and chimpanzees alongside high alertness suggests that the association between ingroup cohesion and outgroup cues may predate the divergence of bonobos and chimpanzees, decreasing in intensity alongside reduced intergroup competition in bonobos. When considered alongside results in humans (e.g. [46–48]), this suggests the phenomenon may also predate the *Pan-Homo* split. While it remains possible that the association evolved independently after the two lineages diverged, the most parsimonious interpretation of the presently available data is that the association could be found in the last common ancestor of humans and *Pan*. Although further systematic comparisons are needed from wild groups, the current study is consistent with the hypothesis that the common ancestor of humans and *Pan* may have tended to increase group cohesion in response to outgroup threat, potentially as an adaptive response to intergroup competition [1, 49], and therefore provides some tentative and preliminary evidence for intergroup competition in the last common Pan-Homo ancestor. Interestingly, patterns of intergroup relations vary considerably between species of gorillas, from more competitive and territorial mountain gorillas (*Gorilla*

*beringei*) to more tolerant western lowland gorillas (*G. gorilla*) [50, 51]. Some evidence suggests the association between outgroup threat and ingroup cohesion can be found in mountain gorillas [15], and thus if the same pattern is found in western lowland gorillas it would point towards an even earlier emergence of this phenomenon in hominids. On the other hand, decreased social cohesion following intergroup cues (either real or simulated) in tufted capuchin monkeys (*Cebus apella*) [16], bonnet macaques (*Macaca radiata*) [14], and lion-tailed macaques (*Macaca silenus*) [18] suggests that the pattern is not shared across haplorrhines.

It will also be important to assess the existence of this effect across different grouping structures and other taxa. Most prior work has focused on cooperative breeders with a single breeding pair, and while both bonobos and chimpanzees live in fission-fusion social dynamics with groups containing multiple reproductively active adults of both sexes, this still represents a narrow range of the diversity of animal social organization shaping species' group-based behaviour and cognition. The existence and strength of this effect in non-human species which form multi-level societies would be especially interesting avenue for future work, as aggregations at different structural levels may have different functional importance to members. More tests with systematic variation in species' socio-ecologies in comparative research is essential.

Despite some important limitations and constraints on ecological validity given the captive context, we overall find a pattern consistent with the social cohesion hypothesis for bonobos' proximate response to outgroup vocalizations. This provides evidence that increased ingroup social affiliation in response to outgroup threat may be conserved in the *Pan-Homo* lineage, and suggests the possible presence of intergroup competition (alongside selection for group-based behaviour and cognition) in our last common ancestor.

## Supporting information

**S1 File. This file contains all supporting information for this manuscript including: Visualizations of effects by trial, stability and collinearity models results, singular model results, all code used in analyses and figure creation (as well as saved models), all data, both figures, and two examples each of control and outgroup stimuli used in this experiment.**
(ZIP)

## Acknowledgments

We first thank the bonobos who participated in this study. We also thank Ape Initiative, Cologne Zoo, Frankfurt Zoo, Kumamoto Sanctuary, and Twycross Zoo for their immense support in this study. We are especially grateful to Lisa Gillespie, Jennifer Guebert, Satoshi Hirata, Haley Holmes, Sabrina Linn, Etsuko Nogami, Jess Rendle, Johanna Rode-White, Alexander Sliwa, Anna Thomas, Dalma Zsalako, and the bonobo caretakers at each site for their help coordinating and running the study.

## Author Contributions

**Conceptualization:** James Brooks, Shinya Yamamoto.

**Data curation:** James Brooks.

**Formal analysis:** James Brooks.

**Funding acquisition:** James Brooks, Shinya Yamamoto.

**Investigation:** James Brooks, Karlijn van Heijst, Amanda Epping, Seok Hwan Lee, Aslihan Niksarli, Amy Pope, Zanna Clay, Mariska E. Kret, Jared Taglialatela, Shinya Yamamoto.

**Methodology:** James Brooks, Shinya Yamamoto.

**Project administration:** James Brooks, Zanna Clay, Mariska E. Kret, Jared Taglialatela, Shinya Yamamoto.

**Resources:** James Brooks, Zanna Clay, Mariska E. Kret, Shinya Yamamoto.

**Supervision:** Zanna Clay, Mariska E. Kret, Jared Taglialatela, Shinya Yamamoto.

**Visualization:** James Brooks.

**Writing – original draft:** James Brooks.

**Writing – review & editing:** James Brooks, Karlijn van Heijst, Amanda Epping, Seok Hwan Lee, Aslihan Niksarli, Amy Pope, Zanna Clay, Mariska E. Kret, Jared Taglialatela, Shinya Yamamoto.

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
