## [Decision Letter · Decision Letter 0]

11 Mar 2024

PONE-D-23-33770Outgroup cues promote ingroup cohesion in bonobos despite limited wild intergroup competitionPLOS ONE

Dear Dr. Brooks,

Thank you for submitting your manuscript to PLOS ONE. After careful consideration, we feel that it has merit but does not fully meet PLOS ONE’s publication criteria as it currently stands. Therefore, we invite you to submit a revised version of the manuscript that addresses the points raised during the review process.

We look forward to receiving your revised manuscript.

Kind regards,

Rachael Miller (Harrison)

Academic Editor

PLOS ONE

Journal Requirements:

Did you know that depositing data in a repository is associated with up to a 25% citation advantage (https://doi.org/10.1371/journal.pone.0230416)? If you’ve not already done so, consider depositing your raw data in a repository to ensure your work is read, appreciated and cited by the largest possible audience. You’ll also earn an Accessible Data icon on your published paper if you deposit your data in any participating repository (https://plos.org/open-science/open-data/#accessible-data).

4. Please expand the acronym “FOREST” (as indicated in your financial disclosure) so that it states the name of your funders in full.

5. Please ensure that you refer to Figure 2 in your text as, if accepted, production will need this reference to link the reader to the figure.

**Additional Editor Comments:**

I have received two reviews of your manuscript with several recommendations that lead me to request major revisions in order for your manuscript to be considered for publication in PLOS ONE. Please ensure that you address and/or respond to each recommendation, in particular, I agree the reviewers for the need to tone down the interpretation of the results. 

Reviewers' comments:

Reviewer's Responses to Questions

**Comments to the Author**

1. Is the manuscript technically sound, and do the data support the conclusions?

Reviewer #1: Partly

Reviewer #2: Partly

2. Has the statistical analysis been performed appropriately and rigorously? 

Reviewer #1: Yes

Reviewer #2: Yes

3. Have the authors made all data underlying the findings in their manuscript fully available?

Reviewer #1: Yes

Reviewer #2: Yes

4. Is the manuscript presented in an intelligible fashion and written in standard English?

Reviewer #1: Yes

Reviewer #2: Yes

5. Review Comments to the Author

Reviewer #1: General:

The analyses are well done and the methods are clear, but I cannot recommend this paper for publication without a substantial re-write that tones down claims of increased in group cohesion. The results show that bonobos sat up more during the playback but “all significant effects (except the proportion of rest sitting upright) interacted with trial, suggesting the effect was not uniform or universal.” It takes weak results and makes very bold claims. I would be more interested in reading a realistic account of how the bonobos responded to the playback and that this doesn’t meet the authors’ expectations. I wonder whether the methods accurately reflected bonobo intergroup encounters as no female vocal stimuli were used, and how appropriate it is to simulate intergroup encounters in captive individuals who have lived experience of joining new individuals in an artificial setting. I would like to encourage the authors to resubmit this paper after restructuring it to more accurately represent the findings.

Title: The title is misleading as the results do not show substantial increases in group cohesion during or after playback. It is more appropriate to say that “Captive bonobos are attentive to outgroup cues but with limited effect on ingroup cohesion”

Abstract: The start is a bit vague – why is the evolutionary history unclear? Which species does this occur in? why does adding bonobos make it clearer?

Are outgroup encounters necessarily a “threat”? It might be more appropriate to talk about “outgroup competition” or how the “likelihood of encountering an outgroup” increases group cohesion. They might be high-stress, high-arousal encounters without being considered a threat. How likely are these particular individuals to consider unknown individuals as a threat given their rearing history?

Line 48: What does “groups” mean here for these species? There’s a potential difference for perceived threats between a breeding pair or a large group of multiple adult individuals that includes non-kin and diverse pairings

Paragraph starting line 60: In this section, while you introduce the idea that humans do intergroup conflict, I recommend also indicating that our intergroup relationships may be more bonobo-like than chimp-like. Chimpanzee intergroup relationships are always hostile, but human and bonobo intergroup relationships may be hostile but can also be friendly (and often sit along a spectrum between these extremes). This would more accurately represent human group relationships and strengthens the justification for bringing in a bonobo comparison.

Line 102: I was a bit confused in the wording of the hypothesis and predications, and would suggest re-working this section to make the predictions clearer to the reader e.g. “This hypothesis predicts: (a) increased affiliation, including behaviours like play and groomings; (b) decreased aggression; (c)… Whereas the null hypothesis is…” You can then also then follow this up in the discussion and identify which of these predictions were met and how strong the effect was and give a more balanced view of your evidence for the social cohesion hypothesis.

Methods

Line 111: Is there a supplemental information showing the demographics of each group? And how large are their enclosures, as this may be important to interpreting proximity data?

Line 143: was any training and/or interobserver reliability conducted with the observers? And were the observers aware of or naïve to the hypothesis being tested? Given the decision to do live coding rather than video coding, we need some more information on the observers

Line 163: are the 12 recordings from 12 different adult male bonobos? And then on line 170, are the 4 distinct recordings from the same or four different individuals? I imagine that it’s from four different bonobos to simulate a group encounter but it’s currently a bit unclear.

Why did you choose to only use male bonobo calls rather than a mix of male and female calls? Given that females often lead in intergroup encounters, maybe including female calls in the playback would have produced a different effect. This may have been a choice to provide direct comparison to the chimps, but I worry it might be at the cost of ecological validity.

Analysis section is very clearly explained!

Discussion

I am not convinced that this evidence supports the social cohesion hypothesis and am uncomfortable with that being the conclusion. The big effect was that they sat up more, but there was limited evidence that they behaved in a more cohesive way. The results demonstrate that bonobos are more attentive when there is a playback than when there is no playback, and perhaps that they experience elevated stress. Drawing further theoretical interpretations is problematic. Re-iterating the predictions at the start of the discussion and whether or not each was met and with what effect, would allow you to demonstrate that there was some evidence towards the hypothesis without accepting the hypothesis in its entirety.

Reviewer #2: This paper tests the hypothesis that perceived outgroup threat promotes ingroup cohesion in bonobos. Bonobos are an ideal model species for testing this hypothesis due to their relaxed intergroup dynamics compared to chimpanzees. The authors examine changes in the frequency of social interactions (grooming, playing, aggression) and stress-related behavior (self-directed behavior and sitting upright) after the playback of vocalizations from unfamiliar outgroup conspecifics and control crow vocalizations. The topic is interesting and the paper is a good fit for PLOS ONE. However, I have some major concerns that must be addressed before I could recommend this paper for publication.

The authors seem to have mixed up the proximate and ultimate explanations for sociality in their hypotheses and predictions. The rationale behind the “intergroup competition hypothesis” is that social cohesion is an adaptive response to outgroup threat because individuals have a bigger chance to outcompete rival groups as a cohesive unit, and winning intergroup competition provide fitness benefits. And so in species where outgroup threat is strong and the immediate costs of intergroup competition is high (e.g., have lethal consequences), we would expect increased social cohesion and cooperation within group, especially in the context of intergroup interactions. But for species that experience low levels of out-group threat, like the bonobos, we would expect no effect of outgroup threat on sociality in all conditions across all time points. Therefore, to investigate the effect of hearing outgroup vocalizations on sociality and vigilance behaviors, the authors should compare occurrences of a given behavior before and after playback for both control and outgroup conditions. Yet, the authors only reported results during and after playback, making it hard to assess whether behavioral patterns change relative to baseline in the two conditions. Rather than being a separate hypothesis, the “social cohesion hypothesis” seems to be predictions derived from the “intergroup competition hypothesis”. Also, can the authors elaborate on the “conserved association hypothesis”? In the absence of strong outgroup threat and competition, what may be the selection pressure for the evolution of group cohesion and cooperation in bonobos?

While proximity and aggression are relevant measures of group cohesion, grooming and play are more measures of affiliation. Accordingly, I disagree with the authors on the link between outgroup threat and ingroup cohesion in bonobos. Rather, the results on the effect of condition on grooming and vigilance seem to indicate individual stress response to outgroup vocalizations.

I appreciate that the authors examine the proximate consequences of outgroup threat across two different time points, with the "during phase" representing the immediate response to the vocalizations and the "post phase" representing the delayed response. But I disagree with the authors that the consequence of outgroup threat on grooming is “short-lived”. Based on Figure S1, there seems to be a lasting effect of hearing outgroup vocalizations on grooming, as the probability to groom increases with increasing time lapse after hearing outgroup but not control vocalizations.

Related to the above comment, I am concerned whether the control conditions are truly controls on days which outgroup trials precede control trials. I understand that there may be logistics reasons for why control and outgroup trials are conducted on the same day. One way to address this issue is to compare behavioral occurrences before and after playback in each trial.

I also do not understand why the authors are interested in the interaction between trial and condition. Do the authors predict that the potential effect of habituation as trials progress would be different for control and outgroup conditions? If so, why? It seems more intuitive to me to test the main effect of condition on behaviors while accounting for any potential habituation effect by including trial number as an independent control predictor (rather than an interaction term) in all of the models. For the models that examine social behaviors, the authors should also account for the opportunity to interact with another individual by including the number of individuals in the group.

Minor comments:

Line 87: Please define outgroup and control conditions.

Line 156: You may need to include a random effect of day for all the models to account for non-independence of data points of the same experimental day.

Line 160-161: For sites where multiple groups were tested, did the groups respond to the playback vocally? I wonder if behavioral changes post playback may be influenced by the vocal response of the other group.

Line 163: Can you provide more information about the context in which these recordings were taken (e.g., during travelling, feeding, fusion or aggressive events)? Also, are these calls from 12 different male bonobos? If not, you may have to account for any potential random preferences for a specific caller in your models (by including caller ID as random effect).

Line 176: What are these randomly selected sounds? Are they bonobo or crow vocalisations or something completely different and novel?

Line 201-204: Have you inspected whether the fixed effects you included as random slopes are actually identifiable (i.e., vary within levels of the random effect)? You should exclude random slopes that are unidentifiable before dropping identifiable random slopes because excluding identifiable random slopes can lead to Type 1 error (see recommendation from Bates D, Kliegl R, Vasishth S & Baayen H. Parsimonious Mixed Models. https://arxiv.org/abs/1506.04967v1).

Line 222: Please specify the terms included in each null model.

Line 298: I am not sure if all readers are familiar with the term “phylogenetic inertia”. Please provide a definition for that.

Figure 1 and 2: For clarity, can you bin the data points into circles of different sizes, where the size of circles represents the number of dyads at different levels of the response for each proximity category and condition?

Figure 2: Please include patterns of sexual behavior in this figure.

6. PLOS authors have the option to publish the peer review history of their article (what does this mean?). If published, this will include your full peer review and any attached files.

Reviewer #1: No

Reviewer #2: No

---

## [Author Response · Author response to Decision Letter 0]

8 Jun 2024

Reviewer #1: General:

The analyses are well done and the methods are clear, but I cannot recommend this paper for publication without a substantial re-write that tones down claims of increased in group cohesion. The results show that bonobos sat up more during the playback but “all significant effects (except the proportion of rest sitting upright) interacted with trial, suggesting the effect was not uniform or universal.” It takes weak results and makes very bold claims. I would be more interested in reading a realistic account of how the bonobos responded to the playback and that this doesn’t meet the authors’ expectations. I wonder whether the methods accurately reflected bonobo intergroup encounters as no female vocal stimuli were used, and how appropriate it is to simulate intergroup encounters in captive individuals who have lived experience of joining new individuals in an artificial setting. I would like to encourage the authors to resubmit this paper after restructuring it to more accurately represent the findings.

Thank you for your review of our manuscript. We have rephrased and reformatted the main conclusions to take a more moderate tone throughout the text. We agree that the results on anything beyond alertness were mild, but also note that with predefined statistical measures following the prior chimpanzee study we see similar patterns of significant effects, making it more difficult to interpret and requiring more caution than we showed in the first submitted draft. We have rewritten several sections to better reflect the reality of the results found, especially the discussion (and title). We also firmly agree that perhaps in the case of bonobos female vocalizations may have a different or more pronounced effect, and that captive groups would likely have radically different group-related behavioural responses than wild counterparts, but emphasize our goal was to follow the chimpanzee study as closely as possible and as such consider this to be a necessary first step (but add much-needed discussion around this point).

Title: The title is misleading as the results do not show substantial increases in group cohesion during or after playback. It is more appropriate to say that “Captive bonobos are attentive to outgroup cues but with limited effect on ingroup cohesion”

We have changed the title to reflect the results more accurately (line 1).

Abstract: The start is a bit vague – why is the evolutionary history unclear? Which species does this occur in? why does adding bonobos make it clearer?

We have given more detail to the reasoning and importance of bonobos as a study species for this question in the abstract (lines 19-20).

Are outgroup encounters necessarily a “threat”? It might be more appropriate to talk about “outgroup competition” or how the “likelihood of encountering an outgroup” increases group cohesion. They might be high-stress, high-arousal encounters without being considered a threat. How likely are these particular individuals to consider unknown individuals as a threat given their rearing history?

This is an interesting point. The intergroup relations of any real species are inevitably nuanced and the term “threat” may not be universally appropriate here, but at the same time “outgroup threat” itself has been the target of much of the prior literature (especially that in humans and in theoretical studies) and is the phenomenon generally being targeted. We have rephrased at several points to reflect more nuance, especially distinguishing more specific practical information (referring to stimuli and raw effects by perception of outgroup cues only instead of perception of outgroup threat) from more theoretical sentences aiming at the target phenomenon beyond this specific study (where much existing discussion highlights outgroup threat itself). We hope these changes both reliably reflect previous studies and theory but do not overgeneralize this context to any/all intergroup relations.

Line 48: What does “groups” mean here for these species? There’s a potential difference for perceived threats between a breeding pair or a large group of multiple adult individuals that includes non-kin and diverse pairings

This is also an interesting point and one we come back to in the discussion (lines 434-442). In the introduction we rephrase to highlight simply that group structures vary significantly and may not always be representing the same thing (lines 50-56), and in the discussion come back and highlight the importance of this diversity as a particular area of study, especially as it relates to species which form multi-level societies (lines 434-442).

Paragraph starting line 60: In this section, while you introduce the idea that humans do intergroup conflict, I recommend also indicating that our intergroup relationships may be more bonobo-like than chimp-like. Chimpanzee intergroup relationships are always hostile, but human and bonobo intergroup relationships may be hostile but can also be friendly (and often sit along a spectrum between these extremes). This would more accurately represent human group relationships and strengthens the justification for bringing in a bonobo comparison.

This is important, we have rephrased (lines 67-69).

Line 102: I was a bit confused in the wording of the hypothesis and predications, and would suggest re-working this section to make the predictions clearer to the reader e.g. “This hypothesis predicts: (a) increased affiliation, including behaviours like play and groomings; (b) decreased aggression; (c)… Whereas the null hypothesis is…” You can then also then follow this up in the discussion and identify which of these predictions were met and how strong the effect was and give a more balanced view of your evidence for the social cohesion hypothesis.

According to your and reviewer 2’s suggestions, we have rewritten this section entirely aiming to clarify the structure of the hypotheses and predictions as well as their theoretical motivation (lines 86-111). We also take your advice to use letters to indicate subpredictions and then follow this structure in reviewing the those that were met in the results at the beginning of the discussion section (lines 309-320).

Methods

Line 111: Is there a supplemental information showing the demographics of each group? And how large are their enclosures, as this may be important to interpreting proximity data?

We include a sheet title “IndividualInfo.xlsx” in the data subfolder of our supplementary file which indicates the name, sex, and age of participants at all sites/groups. While we cannot list the exact sizes of all enclosures, we add a statement that they were comparable (or often larger) than the chimpanzee enclosures in the prior study, suggesting ceiling effects cannot explain the lack of effect in bonobos (lines 143-144).

Line 143: was any training and/or interobserver reliability conducted with the observers? And were the observers aware of or naïve to the hypothesis being tested? Given the decision to do live coding rather than video coding, we need some more information on the observers

We have added more detail about observer training (lines 164-167). Given the varied enclosures we found video-recording of all settings to be impossible, but aimed to use only measures that were relatively direct to observe and which are employed in field observation where similar challenges apply. At each site observers trained together until reaching a high level of consistency, but given practical constraints this was not systematically assessed, and observers were aware of hypotheses. We specify these limitations more explicitly.

Line 163: are the 12 recordings from 12 different adult male bonobos? And then on line 170, are the 4 distinct recordings from the same or four different individuals? I imagine that it’s from four different bonobos to simulate a group encounter but it’s currently a bit unclear.

Why did you choose to only use male bonobo calls rather than a mix of male and female calls? Given that females often lead in intergroup encounters, maybe including female calls in the playback would have produced a different effect. This may have been a choice to provide direct comparison to the chimps, but I worry it might be at the cost of ecological validity.

Yes, the 12 recordings are from 12 different adult males. The 4 distinct recordings are from four different individuals in all cases. We specify this more clearly (lines 189-198).

It was indeed a difficult choice for us to decide whether to use male or female vocalizations, or some combination of both. We considered that for bonobo groups, females would be arguably more central to intergroup relations, but also that aggression during intergroup encounters still often comes from males and thus unfamiliar males may be perceived as more threatening. Further, we considered consistency with the prior chimpanzee study to be the highest priority, and as such that the most straightforward approach was keeping sex consistent. That said, we recognize and agree that there seems to be some trade-off between consistency and ecological validity here. We believe that for this initial study of bonobos, male vocalizations provided the most direct test of the core hypotheses, but now discuss this trade-off in the discussion (lines 402-412).

Analysis section is very clearly explained!

Thank you!

Discussion

I am not convinced that this evidence supports the social cohesion hypothesis and am uncomfortable with that being the conclusion. The big effect was that they sat up more, but there was limited evidence that they behaved in a more cohesive way. The results demonstrate that bonobos are more attentive when there is a playback than when there is no playback, and perhaps that they experience elevated stress. Drawing further theoretical interpretations is problematic. Re-iterating the predictions at the start of the discussion and whether or not each was met and with what effect, would allow you to demonstrate that there was some evidence towards the hypothesis without accepting the hypothesis in its entirety.

This is a fair point, and we concur the submitted draft overstated the findings. The results were quite moderate. As you state, we find some but not complete evidence for the hypothesis, and must convey the evidence accurately and discuss what it suggests for the broader field without overstating its implications. As you recognize, we similarly could not feel comfortable rejecting the social cohesion hypothesis entirely and endorsing the recent intergroup competition hypothesis, as with predefined statistical approaches we did see a very similar pattern emerge as seen in the chimpanzee study. The data requires a more subtle interpretation. We follow your suggestion to first restate the alternate predictions against the data to reflect this (related to the restructuring of the hypotheses as well to aid this discussion). We also tone down our overall conclusion, are careful to give more balanced phrasing to any statements, and remove some sentences/paragraphs entirely which overstated the results. We still hold that the most parsimonious explanation for the pattern observed is a weak, but not absent, effect in bonobos, potentially conserved from a common ancestor but no longer under selective pressure, but we aim to discuss this without giving readers a misconception that this is a proven fact. We are therefore more careful not to frame this as a firm conclusion but as the current hypothesis for which the data is most consistent, with the important caveats we raise throughout the discussion, and as tentative evidence but not proof of any hypothesis (throughout the manuscript, but especially lines 309-332, 413-433, and 443-448).

Reviewer #2: This paper tests the hypothesis that perceived outgroup threat promotes ingroup cohesion in bonobos. Bonobos are an ideal model species for testing this hypothesis due to their relaxed intergroup dynamics compared to chimpanzees. The authors examine changes in the frequency of social interactions (grooming, playing, aggression) and stress-related behavior (self-directed behavior and sitting upright) after the playback of vocalizations from unfamiliar outgroup conspecifics and control crow vocalizations. The topic is interesting and the paper is a good fit for PLOS ONE. However, I have some major concerns that must be addressed before I could recommend this paper for publication.

The authors seem to have mixed up the proximate and ultimate explanations for sociality in their hypotheses and predictions. The rationale behind the “intergroup competition hypothesis” is that social cohesion is an adaptive response to outgroup threat because individuals have a bigger chance to outcompete rival groups as a cohesive unit, and winning intergroup competition provide fitness benefits. And so in species where outgroup threat is strong and the immediate costs of intergroup competition is high (e.g., have lethal consequences), we would expect increased social cohesion and cooperation within group, especially in the context of intergroup interactions. But for species that experience low levels of out-group threat, like the bonobos, we would expect no effect of outgroup threat on sociality in all conditions across all time points. Therefore, to investigate the effect of hearing outgroup vocalizations on sociality and vigilance behaviors, the authors should compare occurrences of a given behavior before and after playback for both control and outgroup conditions. Yet, the authors only reported results during and after playback, making it hard to assess whether behavioral patterns change relative to baseline in the two conditions. Rather than being a separate hypothesis, the “social cohesion hypothesis” seems to be predictions derived from the “intergroup competition hypothesis”. Also, can the authors elaborate on the “conserved association hypothesis”? In the absence of strong outgroup threat and competition, what may be the selection pressure for the evolution of group cohesion and cooperation in bonobos?

Thank you for your detailed review of our manuscript and suggestions for its improvements. We found the discussion of hypotheses especially helpful, as this was not well-stated in our earlier version, and we believe has been significantly improved thanks to your review.

As you suggest, the social cohesion hypothesis details the proximate predictions, but derives from broader ultimate hypotheses. Specifically, there are two alternate ultimate hypotheses (conserved association and intergroup competition) for the existence of this effect in bonobos, which yield different proximate hypotheses (social cohesion and null, respectively). We now explicitly distinguish these levels of analysis and the hypotheses and predictions that relate to them, as well as list more clearly the behavioural effects and categories for the predictions of each as suggested by reviewer 1 (lines 82-111). We hope this clears up any confusion and frames our study in a more theoretical coherent manner where the logic was implicit or messy.

Related to your point about the experimental design, there are two main reasons we did not directly compare behaviour before and after playback in each trial. The first of which is that we wanted to keep analyses identical to the chimpanzee study, and the second of which is, relatedly, that we do not consider the before and during phases to be as reflective of the hypothesis to be tested as such (here or in the chimpanzee study). More specifically, there was in some groups feedings that occurred prior to the “pre” phase. In these groups we ensured the same feeding occurred in all trials and in both the morning and afternoon sessions, and that there was no intervention or changes to their condition at any point during the observation (including the 30 minute pre phase), but we note that the pre phase therefore included gradual depletion of their foods and was a different social context than the during phase (beyond the stimulus presentations). Our primary study design was comparing outgroup to control condition, so although we observed prior to the during phase because we wanted to be sure there had not been unusual events, the specific behavioural data here is no

---

## [Decision Letter · Decision Letter 1]

16 Jul 2024

Increased alertness and moderate ingroup cohesion in bonobos’ response to outgroup cues

PONE-D-23-33770R1

Dear Dr. Brooks,

We’re pleased to inform you that your manuscript has been judged scientifically suitable for publication and will be formally accepted for publication once it meets all outstanding technical requirements.

Kind regards,

Rachael Miller (Harrison)

Academic Editor

PLOS ONE

Additional Editor Comments (optional):

Thank you for your careful revision of the manuscript in line with the suggestions and comments from the two reviewers. One of the reviewers kindly agreed to re-review the re-submission, please see their comments below. I am happy to accept the submission at this point. 

Reviewers' comments:

Reviewer's Responses to Questions

**Comments to the Author**

1. If the authors have adequately addressed your comments raised in a previous round of review and you feel that this manuscript is now acceptable for publication, you may indicate that here to bypass the “Comments to the Author” section, enter your conflict of interest statement in the “Confidential to Editor” section, and submit your "Accept" recommendation.

Reviewer #2: All comments have been addressed

2. Is the manuscript technically sound, and do the data support the conclusions?

Reviewer #2: Yes

3. Has the statistical analysis been performed appropriately and rigorously? 

Reviewer #2: Yes

4. Have the authors made all data underlying the findings in their manuscript fully available?

Reviewer #2: Yes

5. Is the manuscript presented in an intelligible fashion and written in standard English?

Reviewer #2: Yes

6. Review Comments to the Author

Reviewer #2: This revised submission is substantially improved from the original manuscript, mainly in clarifying the hypotheses and toning down the conclusion. I am excited to see this work published in PLOS ONE.

Even though the authors did not consider the suggested changes on the analyses, I understand that it is to provide direct comparison to their chimpanzee study and acknowledge the value in that. I appreciate all the clarifications on the experimental design. I also think the point about potential effect of food provision in the pre phase on bonobo behavior is very valid and should be mentioned in the Discussion as a potential confound. I also appreciate the toning down of the conclusion and result interpretations, especially with rephrasing outgroup vocalizations as an outgroup cue rather than outgroup threat as it is unclear whether the bonobos perceive these calls as a threat based on their behavioral response. Finally, it is good to see that the authors acknowledged the limitation of only using male bonobo vocalizations as outgroup cues in this study. I think it is an important point given the key role females play in bonobo intergroup encounters in the wild. And with sex-related attention biases shown in eye-tracking studies, it may be that male and female bonobos respond differently to outgroup cues – a potential avenue for future research.

7. PLOS authors have the option to publish the peer review history of their article (what does this mean?). If published, this will include your full peer review and any attached files.

Reviewer #2: No

---

## [Editor Report · Acceptance letter]

25 Jul 2024

PONE-D-23-33770R1 

PLOS ONE

Dear Dr. Brooks, 

I'm pleased to inform you that your manuscript has been deemed suitable for publication in PLOS ONE. Congratulations! Your manuscript is now being handed over to our production team.

Kind regards, 

on behalf of

Dr. Rachael Miller (Harrison) 

Academic Editor

PLOS ONE